# Multicenter Experience with Good Manufacturing Practice Production of [^11^C]PiB for Amyloid Positron Emission Tomography Imaging

**DOI:** 10.3390/ph17020217

**Published:** 2024-02-07

**Authors:** Anders Bruhn Arndal Andersen, Szabolcs Lehel, Ebbe Klit Grove, Niels Langkjaer, Dan Fuglø, Tri Hien Viet Huynh

**Affiliations:** 1Department of Nuclear Medicine, Copenhagen University Hospital, Herlev and Gentofte, Borgmester Ib Juuls Vej 1, 2730 Herlev, Denmark; anders.bruhn.arndal.andersen@regionh.dk (A.B.A.A.); dan.fugloe@regionh.dk (D.F.); 2Department of Clinical Physiology, Nuclear Medicine and PET, Copenhagen University Hospital Rigshospitalet, 2100 Copenhagen, Denmark; szabolcs.lehel@regionh.dk; 3Department of Nuclear Medicine and PET Centre, Aarhus University Hospital, 8200 Aarhus, Denmark; ebbgro@rm.dk; 4Department of Nuclear Medicine, Odense University Hospital, 5000 Odense, Denmark; niels.langkjaer@rsyd.dk

**Keywords:** [^11^C]PiB, radiosynthesis, PET/CT imaging, GMP production, Alzheimer’s disease

## Abstract

Alzheimer’s disease (AD) is a neurodegenerative disorder with increasing global prevalence and accounts for over half of all dementia cases. Early diagnosis is paramount for not only the management of the disease, but also for the development of new AD treatments. The current golden standard for diagnosis is performed by positron emission tomography (PET) scans with the tracer [^11^C]Pittsburg Compound B ([^11^C]PiB), which targets amyloid beta protein (A*β*) that builds up as plaques in the brain of AD patients. The increasing demand for AD diagnostics is in turn expected to drive an increase in [^11^C]PiB-PET scans and the setup of new [^11^C]PiB production lines at PET centers globally. Here, we present the [^11^C]PiB production setups, experiences, and use from four Danish PET facilities and discuss the challenges and potential pitfalls of [^11^C]PiB production. We report on the [^11^C]PiB production performed with the 6-OH-BTA-0 precursor dissolved in either dry acetone or 2-butanone and by using either [^11^C]CO_2_ or [^11^C]CH_4_ as ^11^C- precursors on three different commercial synthesis modules: TracerLab FX C Pro, ScanSys, or TracerMaker. It was found that the [^11^C]CO_2_ method gives the highest radioactive yield (1.5 to 3.2 GBq vs. 0.8 ± 0.3 GBq), while the highest molar activity (98.0 ± 61.4 GBq/μmol vs. 21.2 to 95.6 GBq/μmol) was achieved using [^11^C]CH_4_. [^11^C]PiB production with [^11^C]CO_2_ on a TracerLab FX C Pro offered the most desirable results, with the highest yield of 3.17 ± 1.20 GBq and good molar activity of 95.6 ± 44.2 GBq/μmol. Moreover, all reported methods produced [^11^C]PiB in quantities suitable for clinical applications, thus providing a foundation for other PET facilities seeking to establish their own [^11^C]PiB production.

## 1. Introduction

The radiopharmaceutical 2-(4′-*N*-[^11^C]methylaminophenyl)-6-hydroxybenzothiazole ([^11^C]6-OH-BTA-1), also known as [^11^C]Pittsburg Compound B ([^11^C]PiB, Figure 1), is used for early diagnosis of Alzheimer’s disease (AD) by positron emission tomography (PET) [1,2]. AD is the most prevalent form of dementia and is characterized by the accumulation of amyloid beta protein (A*β*) as plaques in the brain and acts as the biological target for [^11^C]PiB [2]. The accumulation of A*β* may begin up to two decades before the patient displays AD symptoms and is detectable by PET [2,3,4,5]; thus, [^11^C]PiB-PET is a powerful tool for the differentiation of dementia types, early detection of AD, and assessment of new AD treatments. Recently, immunotherapies that target A*β* have been approved by the FDA, and more candidate drugs are in clinical trials [6]. This will likely drive an increased demand of the early identification of amyloid burden along with a need for increased amyloid-PET availability.

The first ^11^C labeled thioflavin-T derivates were reported [7] in 2002 in the search for potential PET tracers with affinity towards A*β* and shortly after led to the development [1,8] of [^11^C]PiB for A*β* imaging in humans; however, one major challenge of [^11^C]PiB is the short half-life of ^11^C (*t*_½_ = 20.34 min), which in turn requires [^11^C]PiB production to be performed in situ as well as the availability of a cyclotron [9,10]. Moreover, the two primary cyclotron-generated ^11^C-precursors are obtained in the form of either ^11^C-carbondixoide ([^11^C]CO_2_) or ^11^C-methane ([^11^C]CH_4_). They are produced by the proton bombardment of ^14^N_2_ gas (^14^N(p,α)^11^C nuclear reaction) with 0.5–1.0% O_2_ or 5–10% H_2_, respectively [10]. Both [^11^C]CO_2_ and [^11^C]CH_4_ have relatively low chemical reactivities and are therefore often converted into more reactive secondary ^11^C-precursors, such as ^11^C-methyl iodide ([^11^C]CH_3_I) or ^11^C-methyl triflate ([^11^C]CH_3_OTf), which in turn increases the total synthesis time [10]. Fortunately, both [^11^C]CO_2_ and [^11^C]CH_4_ can be produced in good yields, and the main challenge for radiochemists lies in the optimization of the synthesis procedure to obtain ^11^C-products with high radiochemical purity and molar activities (*A*_m_) [9]. The latter is especially desirable [9], as a low *A*_m_ leads to an increased quantity of cold tracers. This may competitively bind to the biological target in question, thereby resulting in the degraded quality of the PET images. Moreover, for brain PET scans the density of receptors is usually low compared to the rest of the body, and a high *A*_m_ becomes instrumental for the visualization [9]. Factors that can contribute to a low *A*_m_ are the sources of 1^2^C presence during ^11^C-tracer synthesis. These can originate from atmospheric [^12^C]CO_2_, the synthesis equipment itself (e.g., tubing, vacuum seals, and vacuum grease), or impurities in reactants, or solvents [9]. Conversely, a high *A*_m_ may, however, lead to the increased radiolytic decomposition of the product, which is also prevalent in ^11^C-chemistry [11]. In such cases, the use of radical scavengers like ascorbic acid may suppress the decomposition process [11] due to its antioxidant properties [12]; they are sometimes utilized in [^11^C]PiB synthesis [13]. Consequently, both radioactive yields and *A*_m_ at the end-of-synthesis (EOS) for ^11^C-radiopharmaceuticals are comprehensive, as these are both unique to the ^11^C-radiotracer in question and greatly influenced by the specific synthesis setups at the individual production sites [9].

In the original radiosynthesis of [^11^C]PiB, *N*-methylation was performed on an *O*-methoxymethyl (MOM) protected precursor, 6-OMOM-BTA-0, by the use of [^11^C]CH_3_I followed by the deprotection of the MOM group with hydrochloric acid in methanol [8]. The radioactive yield at EOS were not reported; however, the reaction gave a high *A*_m_ of 85 GBq/μmol [8]. Today, the most common route is a one-step synthesis by a reaction with [^11^C]CH_3_OTf and the unprotected precursor (6-OH-BTA-0), dissolved in either acetone [14] or 2-butanone [15], as depicted in Figure 1. This method is utilized in the present work. Here, the radioactive [^11^C]CH_3_OTf precursor is obtained by the “gas-method” via either of the two pathways [16,17,18]. These depends on whether [^11^C]CO_2_ (path 1) or [^11^C]CH_4_ (path 2) is produced in the cyclotron target, c.f. Figure 1. In pathway 1, [^11^C]CO_2_ is converted to [^11^C]CH_4_ by a catalytic rection with nickel (Ni) and hydrogen (H_2_) gas at above 300 °C. The [^11^C]CH_4_ (obtained by cyclotron or path 1) is reacted with iodine (I_2_) at above 700 °C to yield [^11^C]CH_3_I, before being converted to [^11^C]CH_3_OTf by a reaction with silver triflate (AgOTf) at 200 °C. Recently, one-pot strategies have also been reported and involve the direct fixation-reduction of cyclotron-produced [^11^C]CO_2_ with 6-OH-BTA-0 the by use of catalytic mixtures of either PhSiH_3_/IPr/ZnCl_2_ [19] or PhSiH_3_/TBAF [20]. Here, the latter approach was able to achieve yields of 9.88 ± 0.59 GBq at EOS and *A*_m_ of 61.4 ± 0.8 GBq/μmol (*n* = 3) [20]; however, these procedures require the preparation of reagent solutions under inert atmosphere prior to [^11^C]PiB production, which may not be practically feasible at clinical sites. For additional information, the reader is referred to Refs. [19,20].

The short *t*_½_ for ^11^C also poses a challenge for the separation and formulation of [^11^C]PiB from the reaction mixture, where fast purification is desirable. Here, the most common strategy has been semi-preparative high performance liquid chromatography (HPLC) [5,13,14,15,20,21], where compound separation is performed by larger HPLC columns and faster flowrates compared to analytical HPLC methods. Various semi-preparative HPLC setups (columns, eluents, and flowrates) have been reported but all with general retention times of less than 10 min; however, in some cases, an additional purification step of the product fraction is performed by elution through commercially available single-use chromatographic cartridges [20,21]. Consequently, a cartridge-only-based separation method has been reported [22]. Here, the precursor was dissolved in acetone and loaded onto the cartridge before reacted with [^11^C]CH_3_OTf as well as [^11^C]PiB eluted with 50% aqueous ethanol [22]. This approach may pave the way for future cassette-based production methods of [^11^C]PiB [22], which currently are not available.

The short *t*_½_ of ^11^C has also led to the development of ^18^F-tracers for A*β* imaging, such as florbetaben [23], florbetapir [24], flutemetamol (“[^18^F]PiB”) [25], and NAV4694 [26], as the *t*_½_ for ^18^F is 109.77 min. Florbetapir was the first fluorinated A*β* tracer and has comparable retention ratios to [^11^C]PiB [2], whereas flutemetamol is now a commercialized A*β* tracer. Moreover, the ^18^F-tracers show no apparent differences in diagnostic accuracy toA*β* tracers [27]. However, the ^18^F-tracers have increased levels of non-specific uptake in the white matter of the brain due to their more lipophilic nature [2]. This results in increased background noise and the potential loss of information on cortical A*β* [2]. Nonetheless, [^11^C]PiB remains the most extensively studied PET tracer for A*β* imaging and has therefore, to our knowledge, become the most widely implemented tracer for this purpose.

In this study, we report and compare recent [^11^C]PiB production results from four clinical sites in Denmark at Aarhus University Hospital (AUH) (production quantity, *n* = 74), Herlev University Hospital (HUH) (*n* = 104), Odense University Hospital (OUH) (*n* = 84), and Rigshospitalet (RH) (*n* = 252), and discuss the advantages and disadvantages of the different synthetic routes and manufacturing approaches employed at the four sites. The production processes are performed on three different automatic “gas-method” synthesis modules for ^11^C-chemisty: ScanSys (software version 1.0) and TracerMaker (software version 1.0) (both from Scansys Laboratorieteknik ApS, Vaerloese, Denmark), and TracerLab FX C Pro (software version 2.2.2,GE HealthCare, Uppsala, Sweden). The loop method (converting [^11^C]CH_4_ to [^11^C]CH_3_I) was developed by Scansys Laboratorieteknik ApS in collaboration with GE HealthCare. GE HealthCare patented the method for 20 years. Scansys Laboratorieteknik ApS implemented the method on a new synthesis module, the TracerMaker, when the patent expired. These three synthesis modules were the only ones with this configuration at the time. Moreover, the clinical use of [^11^C]PiB at the four sites will also be presented. Thus, a starting point for other PET facilities seeking to produce [^11^C]PiB is provided.

## 2. Results

The main parameters and results for the different [^11^C]PiB production processes are summarized in Table 1, and the radioactive yields and *A*_m_ at EOS are shown in Figure 2 for each of the four sites. Detailed production and product separation procedures for each of the four sites are found in the Appendix A.

Generally, AUH, OUH, and RH use [^11^C]CO_2,_ while HUH uses [^11^C]CH_4_, as the ^11^C-precusors for the [^11^C]PiB production, c.f. Figure 1. A production process with [^11^C]CH_4_ is possible at AUH and RH but is not used (vide infra), whereas the cyclotron setup at HUH only allows for production of [^11^C]CH_4_. Overall, the production of [^11^C]PiB for the four sites are as follows: At AUH, OUH and RH, [^11^C]CO_2_ is converted to [^11^C]CH_4_ via a catalytic reaction with Ni and H_2_ at either 350 °C (OUH and RH) or 375 °C (AUH). The [^11^C]CH_4_ is then, regardless of how it is obtained (either from [^11^C]CO_2_ or directly from the cyclotron), cycled through a loop and converted to [^11^C]CH_3_I. The loop consists of a HayeSep trap, an I_2_ oven, and a quartz oven. The HayeSep serves to trap [^11^C]CH_4_ at −160 °C (RH), −130 °C (AUH), −125 °C (HUH), or −100 °C (OUH) before it is released into the loop by heating. From here, a reaction [16,17] with gaseous I_2_ occurs in the quartz oven at either 700 °C (OUH), 720 °C (HUH), 740 °C (AUH), or 760 °C (RH). To help produce I_2_ vapor for the gas-phase reaction, the sublimation of I_2_ crystals is performed by the I_2_ oven at different temperatures dependent on the site in question, c.f. Table 1. Then, the [^11^C]CH_3_I is released and passed through an AgOTf column and heated to 190 °C (AUH) or 200 °C (OUH, HUH, and RH). Here, it is converted [18] to the more reactive [^11^C]CH_3_OTf and subsequently trapped in the reaction vial with the precursor (6-OH-BTA-0) solution at trapping temperatures of −20 °C (AUH), −10 °C (HUH, RH), or 5 °C (OUH). Prior to synthesis, 6-OH-BTA-0 was dissolved in either acetone (AUH and OUH) or 2-butanone (HUH and RH) based on the procedures in the respective original study [14,15] using solvent volumes of 150 μL (HUH) or 300 μL (AUH, OUH, and RH). The [^11^C]PiB reaction is performed at temperatures around the normal boiling point (*T*_N_) of the solvent, i.e., between 50 °C and 60 °C (AUH and OUH) for acetone (*T*_N_: 56.08 °C) and between 80 °C and 85 °C (RH and HUH) for 2-butanone (*T*_N_: 79.64 °C). The reaction times are kept short, i.e., between 2 and 3 min (Table 1), due to the fast methylation reaction rates but also to accommodate for the short *t*_½_ of ^11^C. Moreover, at AUH, the reaction is performed in an external reaction vessel as this gives more consistent yields compared to the internal reactor of the synthesis equipment. The suspected reason is poor cleaning and the drying of the internal reactor and tubing prior to synthesis during the automated production sequence.

The reaction mixture is then loaded onto a semi-preparative HPLC in order to separate [^11^C]PiB into the final product formulation. The equipment and parameters for the product separation are summarized in Table 2. To obtain the semi-preparative conditions, i.e., purification on the milligram scale, large columns (Ø = 10 mm, except HUH) and fast flowrates (6–8 mL/min) are used. This also allows for the relatively fast (>8 min) purification of [^11^C]PiB for all sites, which is important due to the short *t*_½_ of ^11^C. Additionally, at AUH, the product formulation is passed over a SepPak C18 cartridge to prevent excess ethanol in the final product formulation (*vide infra*). For all sites, the collected [^11^C]PiB fraction is transferred via sterile filtration into a sterile product vial with either a phosphate-buffered solution (PBS), as performed at HUH, OUH, and RH, or isotonic saline with 9.1% ethanol, as performed at AUH.

This overall approach affords average yields (and an average *A*_m_) at EOS of 1.98 ± 1.0 GBq (21.2 ± 16.8 GBq/μmol), 0.83 ± 0.29 GBq (98.0 ± 61.4 GBq/μmol), 3.17 ± 1.2 GBq (95.6 ± 44.2 GBq/μmol), and 1.46 ± 0.64 GBq (55.0 ± 50.4 GBq/μmol) for AUH, HUH, OUH, and RH, respectively.

## 3. Discussion

### 3.1. Production and Separation of [^11^C]PiB

#### 3.1.1. [^11^C]CO_2_ vs. [^11^C]CH_4_ Methods

Evidently, the [^11^C]CO_2_ method produces significantly higher radioactive [^11^C]PiB yields compared to the [^11^C]CH_4_ method (Table 1). This reflects differences in the cyclotron production of ^11^C, where the density reduction of the target gas by beam heating depends on the target gas composition and also differences in the interactions of the nucleogenic hot ^11^C atom with the target walls [28]. It has been experimentally shown that [^11^C]CH_4_ yields directly after bombardment are approx. 65% of the [^11^C]CO_2_ yields [28]; however, this is highly dependent on the cyclotron target material, geometry, and cooling efficiency, as well as beam current and irradiation times [9,28]. The reader is referred to refs. [9,28] for more information. In present work, the radioactive [^11^C]PiB yields by using [^11^C]CO_2_ are on average 2.5 times larger at EOS compared to when the [^11^C]CH_4_ method is used. Contrary to this, the [^11^C]PiB molar activities for the [^11^C]CH_4_ method are approx. 1.7 times higher compared to those of [^11^C]CO_2_ (based on the average values for each). This is expected as the catalytic conversion of [^11^C]CO_2_ to [^11^C]CH_4_ (Figure 1) is vulnerable to even small contaminations of atmospheric [^12^C]CO_2_ and poses as the most obvious source of ^12^C, which lowers the *A*_m_ of the final product. To limit the diffusion of [^12^C]CO_2_ into the system, stainless steel lines are used to carry the ^11^C-precursor from the cyclotron to the hot cell at all four production sites, albeit other ^12^C sources also exist, such as the tubing and fittings of the synthesis equipment [9]. From Figure 2, such sources seem to be present for the [^11^C]PiB production process at HUH, as the average *A*_m_ abruptly decrease from an average value of 138.2 ± 60.9 GBq/μmol (*n* = 53) to 56.2 ± 20.0 GBq/μmol (*n* = 51) between the periods of 2019 and 2020, and 2021 and 2023, respectively. These sources are currently under investigation.

#### 3.1.2. Radiolysis Caused by High A_m_

The resultant high *A*_m_ with cyclotron-produced [^11^C]CH_4_ is often desirable in brain PET scans (vide supra); however, it can lead to the radiolytic decomposition of the product. This was observed for the [^11^C]PiB production process at AUH, HUH, and RH. For AUH and RH, this drastically decreased the radioactive yields at EOS compared to the [^11^C]CO_2_ method; even down to a tenth was observed at RH. Moreover, the obtained radiochemical purities were <95% for HUH and RH, i.e., below the acceptance criterion for radiochemical purity (see Appendix A for product specifications and an analysis of all sites). Note: This criterion is deduced from the Ph. Eur. 9.0 monograph #1924: Raclopride ([^11^C]Methoxy) injection and #0125 radiopharmaceutical preparations. This was evident from the radio-trace of the semi-preparative HPLC where the much lower [^11^C]PIB conversion was clearly seen on the radio chromatograms. An example is given for RH in the Appendix A. Thus, [^11^C]CO_2_ for [^11^C]PiB production was chosen over [^11^C]CH_4_ at AUH and RH to negate the issue. Moreover, the GE PETtrace cyclotrons (GE Healthcare, Uppsala, Sweden) at AUH were outfitted with high-pressure [^11^C]CO_2_ targets. This allows for 52.7 μA beam currents and results in significantly higher starting activities at the end-of-bombardment, compared to the 20 μA beam current of the [^11^C]CH_4_ target on the IBA Cyclone 18/18 (IBA, Louvain-la-Neuve, Belgium) used at HUH. Consequently, this made the [^11^C]CO_2_ method more attractive for the [^11^C]PiB production process at AUH. At OUH, [^11^C]CH_4_ production is not available with the current cyclotron setup, and issues with radiolysis were not experienced.

At HUH, the production setup does not readily allow for the use of [^11^C]CO_2_. Thus, other solutions were explored. Here, both *L*-ascorbic acid and ethanol were added as a radical scavengers to the aqueous eluent of the semi-preparative HPLC, to suppress the radiolytic decomposition of [^11^C]PiB. Without these, low yield of [^11^C]PiB was obtained in the final product. The addition of ethanol increased the radiochemical purity up to 68%, and the addition of *L*-ascorbic acid further increased the radiochemical purity of [^11^C]PiB to >95%. Similarly for RH, *L*-ascorbic acid was added to the aqueous semi-preparative HPLC eluent, which ensured the stability of [^11^C]PiB during the HPLC separation process, as well as the >95% radiochemical purity in the final product. Moreover, 2-Butanone was added to the semi-preparative HPLC eluent at HUH to promote solubility and further reduce/inhibit the radiolysis of the product during purification.

#### 3.1.3. Radiochemical and Chemical Impurities

Besides impurities from radiolytic decomposition in the crude [^11^C]PiB product, both radiochemical (radioactive) and chemical (non-radioactive) impurities occurred. For radiochemical impurities, the most likely were unreacted [^11^C]CH_3_OTf and [^11^C]CH_3_I, as well as the byproduct [O-Methyl-^11^C]6-MeO-BTA-0 from the less favorable O-methylation reaction of the precursor 6-OH-BTA-0 (Figure 3a) [8]. These were all removed during the semi-preparative HPLC purification of the [^11^C]PiB product. The formation of [^11^C]CH_3_OH was experienced at AUH and HUH, which occurs in the presence of water reacting with [^11^C]CH_3_OTf (Figure 3b). This led to the use of anhydrous acetone (max 0.01% H_2_O) and dried 2-butanone as reaction solvents at AUH and HUH, respectively, to prevent the formation of [^11^C]CH_3_OH. At AUH, anhydrous acetone was bought from VWR chemicals and used directly, whereas the 2-butanone at HUH was dried for at least 3 days by use of molecular sieves (4 Å). The molecular sieves were prepared prior to the solvent drying by heating to 250 °C for 24 h, before being cooled to room temperature. Additionally, the reactor vial at HUH was also “dried” prior to [^11^C]PiB synthesis by washing it with dry 2-butatone before the addition of the precursor solution.

In regard to chemical purity, the most plausible impurities in the [^11^C]PiB preparation processes are (1) non-radioactive [^12^C]PiB and the 6-OH-BTA-0 precursor, and (2) the residual solvents acetone, 2-butanone, and ethanol, the latter as excipient. The established acceptance levels (Appendix A) for the 6-OH-BTA-0 precursor and [^12^C]PiB were based on already defined specifications for the preparation of [^11^C]PiB for human use at Rigshospitalet, Copenhagen; Aarhus University Hospital; the available literature; as well as the Ph. Eur. 9.0 monograph #1924: Raclopride ([^11^C]Methoxy) injection. These are set to ≤0.2 μg/mL of the 6-OH-BTA-0 precursor and ≤1.0 μg/mL of [^12^C]PiB in the final product solution.

#### 3.1.4. Semi-Preparative HPLC Purification

The purification of [^11^C]PiB from the crude reaction mixture into the final product formulation was performed at all sites using semi-preparative HPLC. The separation procedures were also performed with similar parameters (Table 2) and were based on in-house experiences and from the literature [5,13,14,15,21]. The main differences between these procedures are the choice of HPLC column (manufacturer and dimensions) and the ratio between the aqueous and non-aqueous (ethanol) eluent. The aqueous eluent was buffered to a pH = 2.2 by use of H_3_PO_4_ (HUH, OUH, and RH) or NaH_2_PO_4_ (AUH) to increase the solubility of the product. For the chromatographic radio-trace, two radioactive peaks were observed at all production sites (Appendix A), i.e., one broad peak assigned to radiochemical impurities and a narrow [^11^C]PiB peak. Moreover, at AUH, the removal of ethanol in the [^11^C]PiB fraction before sterilization and product formulation was required. This was due to the large concentration (45%) of ethanol in the semi-preparative HPLC eluent, which caused the ethanol content to exceed the specification (Appendix A) of the final product. Thus, to limit the ethanol, the [^11^C]PiB fraction was passed through a SepPak C18 cartridge during the transfer to the product vial. The C18 cartridge was conditioned prior to synthesis first by 10 mL of ethanol followed by 10 mL of sterile water.

#### 3.1.5. AgOTf Column and Sterile Filtration

In the context of successful [^11^C]PiB production, additional crucial steps are required for the preconditioning of the AgOTf column before synthesis and the selection of sterilization filter, which the product is passed through during the formulation process. Poor AgOTf column conditioning, especially after (re)packing with new column material (AgOTf and Carbopack), causes the insufficient conversion of [^11^C]CH_3_I to [^11^C]CH_3_OTf. This heavily impacts the [^11^C]PiB production process due to the low reactivity of [^11^C]CH_3_I with 6-OH-BTA-0. The conditioning additionally serves to remove potential impurities after column (re)packing. Detailed information on the first-time conditioning and subsequent conditioning prior to each production is found in the Appendix A.

The type of sterilization filter has also been shown to impact the final radioactive yield, as some of the activity may be retained by the filter material [29,30]. This was also observed for the [^11^C]PiB production processes at HUH and OUH. The use of mixed-cellulose-ester (MCE) filter membrane material (Millex-GS 0.22 µm SLGSV255F, Merck, Soeborg, Denmark ) retained 30–40% of the total yield (approx. 400 MBq) at HUH and up to 50% at OUH. Consequently, HUH and OUH tested various sterilization filters based on ref. [30], and the polyvinylidene fluoride (PVDF) filter (Millex-GV 0.22 μm SLGVV255F, Merck, Soeborg, Denmark) showed promising results. For example, approx. 30 Bq (3%) was retained, yielding 1087 MBq [^11^C]PiB at EOS (*n* = 1) at HUH. Currently, HUH is working on getting the PVDF filters approved, whereas the change to PVDF filters has already been implemented at OUH. At AUH and RH, filters with PVDF membrane material were already utilized.

#### 3.1.6. Summary of [^11^C]PiB Production

In summary, the [^11^C]PiB production process requires the careful consideration of multiple parameters which can influence the radioactive yield, *A*_m_, and product purity. This should be considered alongside GMP procedures, and the product specifications required for human use. Even though, the overall production procedures are similar, the choice of ^11^C-precusor, equipment, and work-up procedure have led to site-specific challenges which, in turn, also required site-specific solutions. Particularly, the challenge of avoiding ^12^C-contamination is persistent in the [^11^C]PiB production process of the four production sites. This is especially evident at HUH, with the sudden decrease in *A*_m_ after 2020 due to an unidentified ^12^C-source. Moreover, in the context of the average radioactive yield vs. *A*_m_ at the individual sites, i.e., the choice of ^11^C-precursor ([^11^C]CO_2_ vs. [^11^C]CH_4_), the setup at OUH offers the best compromise. Here, the use of [^11^C]CO_2_ on a TracerLab FX C Pro gives the highest average radioactive yield (3.17 ± 1.20 GBq) of the four sites while also rivaling the best average *A*_m_ obtained by the [^11^C]CH_4_ method at HUH; however, [^11^C]PiB produced on a TracerMaker module with [^11^C]CH_4_ gives the most consistent results in terms of the radioactive yield, as the standard deviation of this parameter is the lowest among the four sites (0.29 GBq).

### 3.2. Clinical Use of [^11^C]PiB

The early diagnosis of neurodegenerative diseases leading to dementia is challenging, and the use of biomarkers like amyloid-PET improves diagnostic accuracy [31]. [^11^C]PiB-PET is used to assess A*β* deposition in grey matter as one of the earliest detectable brain changes in AD; however, A*β* deposition is only weakly correlated to the degree of dementia and must be interpreted in a clinical setting with other biomarkers. [^11^C]PiB PET scans are usually classified as either amyloid-PET-positive or -negative based on visual reading and semi-quantitative analyses. The cerebral/cerebellum SUV_mean_ ratio can be calculated by placing standardized VOIs in cerebral grey matter and using the cerebellar grey matter as reference. At HUH, we use an empirically based SUV_mean_ ratio of 1.5 as the cut-off value. Examples of negative and positive [^11^C]PiB PET scans with corresponding SUV_mean_ ratios are shown in Figure 4.

At HUH, approx. 20 patients were scanned annually in 2019–2021, with an increase to 51 patients in 2022 due to an increased demand. A small number of patients were referred on suspicion of cerebral amyloid angiopathy, while the majority (>95%) were referred on suspicion of AD. These patients are typically referred when clinical findings and other biomarkers (e.g., [^18^F]FDG-PET and CSF-amyloid) are equivocal, in cases of complicating psychological or psychiatric comorbidities that might influence clinical findings, or when patients are young and AD is rare.

Figure 4a shows a [^11^C]PiB-PET scan of a patient with clinical signs of dementia. The type of dementia was difficult to determine clinically, and both frontotemporal dementia and AD were considered. The [^11^C]PiB-PET scan was negative and showed no significant A*β* deposition in the cerebral grey matter. AD was ruled out, and the patient was diagnosed with frontotemporal dementia. The scan in Figure 4b is from a 63-year-old man with clinical signs of AD but with normal CSF-amyloid. Because of the equivocal findings, a [^11^C]PiB-PET scan was performed. The [^11^C]PiB-PET scan was positive with significant A*β* deposition in the cerebral grey matter. The scan result helped the clinicians reach the final diagnosis of AD, and the patient was started on a medical treatment.

At OUH, [^11^C]PiB is being used in a clinical setting to evaluate (younger) patients clinically suspected of AD-type neurodegenerative diseases following a normal or an inconclusive FDG-PET/CT scan. The number of patients and clinical AD indications was 14 in 2020 but increased to 37 and 38 patients in 2021 and 2022, respectively. Moreover, a clinical study is currently being conducted at OUH to describe the glucose metabolism and the formation of amyloid plaques in healthy aging brains. At RH, the number of patients has also gradually increased during years and was 172 in 2022. Here, most patients have the following indications: persistent or progressive unexplained mild cognitive impairment, patients satisfying the core clinical criteria for possible AD because of unclean clinical presentation, either an atypical clinical course or an etiologically mixed presentation, and progressive dementia and an atypical early age of onset (usually defined as 65 years or less in age).

At AUH, the number of patients is on average 25 per year, but [^11^C]PiB has also been used in several studies, for instance, to study the role of cerebrovascular dysfunction in relation to A*β* deposition in preclinical cases of AD. By use of susceptibility contrast MRI (magnetic resonance imaging) and PET, the study has shown evidence of impaired cerebral perfusion with mild cognitive impairment (MCI) in individuals with elevated cortical A*β* (pAD-MCI individuals) relative to MCI individuals without aggregated cortical A*β* (SNAP-individuals). Increased cortical A*β*, in pAD-MCI individuals, correlated with a decreased cerebral blood flow (CBF) as well as an increased mean transit time (MTT) and capillary transit time heterogeneity (CTH) [32].

[^11^C]PiB has been used in a longitudinal study investigating the relationship between levels of inflammation, aggregated A*β* loads, and tau at baseline across a group of MCI cases (predominantly prodromal AD) using PET. This was repeated after a period of two years. From the study, it was found that (a) levels of inflammation decline in prodromal AD cases; (b) MCI cases with low baseline, but subsequently increased amyloid deposition, showed a correlation between levels of inflammation and their A*β* load; and (c) Vice versa, where in MCI cases in which the A*β* load approaches AD levels at baseline, it was shown that the overall levels of inflammation declined over two years. Furthermore, a connection between tau and inflammation levels was found in these subjects [33]. 

Although [^11^C]PiB is commonly used to target the brain, [^11^C]PiB has also been used for targeting the heart in a dual-center study to examine the diagnostic utility of [^11^C]PiB-PET in early cardiac amyloidosis (CA) detection. The study has shown that [^11^C]PiB-PET is an excellent method to distinguish both transthyretin (ATTR) and light-chain (AL) amyloidosis from healthy and hypertrophied controls and has been performed with high accuracy. Furthermore, this method may be able to detect early stages of CA [34].

## 4. Materials and Methods

### 4.1. Radiosynthesis of [^11^C]PiB

#### 4.1.1. AUH

The 6-OH-BTA-0 precursor was obtained from LIMBP, Université P. Verlaine (Metz, France), standard reagents were obtained from Sigma-Aldrich/Merck (Soeborg, Denmark), and sterile solutions were obtained from Aarhus University Hospital Pharmacy (Aarhus, Denmark). Sep-Pak C18 Plus Short cartridges were obtained from Waters (Taastrup, Denmark), and sterile Millex-GV 0.22 µm filters (SLGV033RS) were obtained from Merck Millipore (Soeborg, Denmark), [^11^C]CO_2_ and [^11^C]CH_4_ were obtained by the cyclotron bombardment of the target gas, 99.5% N_2_ + 0.5% O_2_ (Air Liquide Danmark A/S, Taastrup, Denmark) or 95% N_2_ + 5% H_2_ (Air Liquide Denmark A/S, Taastrup, Denmark), respectively, either using a GE PETtrace 600, a GE PETtrace 800, or an IBA Cyclone 18/18 (see Table 3 for parameters). 

Synthesis was carried out on a TracerLab FX C Pro automated synthesis module. The product was isolated by semi-preparative HPLC consisting of a Knauer P 4.1S pump fitted with a Luna C18(2) 5 µm, 250 × 10 mm column (Phenomenex, Broenshoej, Denmark). A detailed description of the [^11^C]PiB radiosynthesis, semi-preparative HPLC, and flow charts is given in the Appendix A.

#### 4.1.2. HUH

The precursor (6-OH-BTA-0) was obtained from ABX advanced biochemical compounds GmbH (Radeberg, Germany), standard reagents were obtained from Sigma-Aldrich/Merk, sterile solutions were obtained from Herlev University Hospital Pharmacy (Herlev, Denmark), and sterile Millex-GS 0.22 µm filters (SLGSV255F) were obtained from Merck Millipore. 2-butanone was dried over molecular sieves (4 Å), and the sieves were heated to 250 °C for 24 h and cooled to room temperature prior to use. [^11^C]CH_4_ was produced using an IBA Cyclone 18/18 cyclotron (Table 3) by the bombardment of the target gas, 95% N_2_ + 5% H_2_ (Air Products).

The automated [^11^C]PiB radiosynthesis and product isolation was performed on a TracerMaker synthesis module (Scansys Laboratorieteknik ApS, Denmark) with a semi-preparative HPLC system with Knauer pumps (AZURA P 2.1S/P 4.1S pumps) and a Kinetex@ 2.6 μm C18 100 Å column (50 × 4.6 mm, Phenomenex). See Table 2 for details on the semi-preparative HPLC eluents and flowrates. See Appendix A for a detailed description of the radiosynthesis and product isolation process.

#### 4.1.3. OUH

The 6-OH-BTA-0 precursor was obtained from either ABX advanced biochemical compounds GTmbH (Radeberg, Germany) or Pharmasynth AS (Tartu, Estonia), standard reagents were obtained from Sigma-Aldrich/Merck, sterile solutions were obtained from either Sygehus Apotek Fyn (Odense, Denmark) or Region Hovedstadens Apotek (Herlev, Denmark), and sterile Cathivex-GV 0.22 µm filters (SLGV0250S) were obtained from Merck Millipore. [^11^C]CO_2_ was produced by the bombardment of the target gas, 99.5% N_2_ + 0.5% O_2_ (Strandmøllen A/S, Klampenborg, Denmark), using a GE PETtrace cyclotron (Table 3).

The automated [^11^C]PiB production and product purification processes were performed on a Tracerlab FXc (GE HealthCare, Uppsala, Sweden) fitted with a S1021 HPLC pump (SYKAM, Eresing, Germany), Chromolith Performance RP-18e column (100 × 10 mm, Merck, Soeborg, Denmark), and K-2001 UV detector (Knauer, Berlin, Germany). The radiosynthesis process and semi-preparative HPLC are described further in Appendix A and Table 2.

#### 4.1.4. RH

The precursor (6-OH-BTA-0) and standard reagents were obtained from ABX advanced biochemical compounds GmbH (Radeberg, Germany), sterile solutions were obtained from Region Hovedstadens Apotek, Denmark), and sterile Millex-GV 0.22 µm filters (SLGV033RS) were obtained from Merck Millipore. [^11^C]CO_2_ was produced using a CTI XP cyclotron on the target gas (99.5% N_2_ + 0.5% O_2_ (Strandmøllen, Klampenborg, Denmark)), and [^11^C]CH_4_ was produced using a Scanditronix cyclotron by the bombardment of the target gas, 95% N_2_ + 5% H_2_ (Strandmøllen, Klampenborg, Denmark). See Table 3 for the parameters.

The automated [^11^C]PiB production process was performed on either a ScanSys or TracerMaker module (Scansys Laboratorieteknik ApS, Vaerloese, Denmark), followed by product isolation by semi-preparative HPLC. For the ScanSys module, the HPLC consists of a Knauer 100 pump, Onyx Monolithic semi-PREP C18 HPLC column (100 × 10 mm, Phenomenex), Knauer UV 120 detector coupled in series with a radioactivity detector, Gilson 401 dilutor, Cavarno XE 1000 syringe pump, and Rheodyne 7750E06 injection valve with a 5 mL loop. For the TracerMaker module, the semi-preparative HPLC system consists of a Knauer P.4.1S pump, Onyx Monolithic semi-PREP C18 HPLC column (100 × 10 mm, Phenomenex), Knauer Azura UVD 215 detector coupled in series with a radioactivity detector, and Rheodyne 7750E06 injection valve with a 5 mL loop. See Appendix A and Table 2 for a detailed description of the radiosynthesis process and product isolation.

### 4.2. Product Quality Control and Analyses

All batches for human use were subjected to quality control tests prior to release in accordance with the respective compassionate use permits issued for each production site by the Danish Medicines Agency. The analytical methods used for QC are generally validated in line with the current European Pharmacopoeia (Ph. Eur. 9.0; monographs #0125 and #1924) and International Conference on Harmonization guidelines. Details on the QC procedures, equipment (dose calibrator, endotoxin test, gas chromatography, HPLC, and pH), and drug specifications for each site are described in the Appendix A.

### 4.3. Statistical Methods

The general data are presented as the average value ± the standard deviation. Data processing was performed in Microsoft Excel 2016 by use of the built-in statistical functionalities and visualized with the Matplotlib package in Python 3.9.

### 4.4. Ethical Treatment of Humans and Animals

The clinical data from all sites were retrieved from the database of existing patient PET/CT scans obtained as part of routine clinical practice. The diagnoses are sufficiently common for specific scans to be discussed without compromising the privacy and confidentiality of the subjects in question. These considerations warrant the absence of ethics committee/review board approval.

## 5. Conclusions

[^11^C]PiB production is reliably performed at four different sites, AUH, OUH, HUH, and RH, by use of different synthesis equipment and variations of the synthesis route. All reported approaches generate [^11^C]PiB in quantities with an *A*_m_ appropriate for clinical use; thus, a foundation for other PET facilities seeking to establish their own GMP compliant [^11^C]PiB production process is provided. The radioactive ^11^C-precursor used is either [^11^C]CO_2_ or [^11^C]CH_4_ and generally impacts the radioactive yield and *A*_m_ of the [^11^C]PiB product at EOS. In the presented work, the use of [^11^C]CO_2_ as the radioactive precursor produces significantly higher radioactive yields (2.5 times higher) compared to [^11^C]CH_4_, whereas the opposite is observed for the *A*_m_ (1.7 times) when [^11^C]CH_4_ is used. Moreover, three different types of synthesis modules are used for [^11^C]PiB production, TracerLab FX C Pro, ScanSys, or TracerMaker, all of which are capable of producing [^11^C]PiB from either of the two ^11^C-precursors and with variations in the synthesis setup, i.e., temperature, solvent (acetone or 2-butanone), and reaction time, as well as the semi-preparative HPLC purification of the [^11^C]PiB product. For the individual average radioactive yields and *A*_m_, the production setup at OUH ([^11^C]CO_2_ on a TracerLab FX C Pro) gives the highest radioactive yield and produces molecular activities close to the average *A*_m_ of the [^11^C]CH_4_ method. The production setup at OUH may therefore be the most desirable method, as it offers the best compromise between the choice of ^11^C-precursors presented in this work.

## Figures and Tables

**Figure 1 pharmaceuticals-17-00217-f001:**
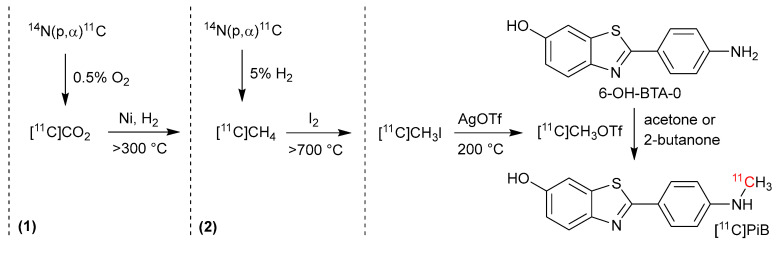
The two different pathways for [^11^C]PiB production discussed in present work by the four different sites: AUH, HUH, OUH, and RH. Pathway (1) is employed at AUH, OUH, and RH, where [^11^C]CO_2_ is produced by ^14^N(p,α)^11^C nuclear reaction in the cyclotron and subsequently converted to [^11^C]CH_4_, [^11^C]CH_3_I, and [^11^C]CH_3_OTf before reacting with 6-OH-BTA-0 precursor. Pathway (2) is used at HUH, where [^11^C]CH_4_ is produced directly by cyclotron ^14^N(p,α)^11^C nuclear reaction, before converted to [^11^C]CH_3_I and [^11^C]CH_3_OTf and reacted with 6-OH-BTA-0. See Table 1 for synthesis parameters, Table 2 for semi-preparative HPLC parameters and Table 3 for cyclotron parameters.

**Figure 2 pharmaceuticals-17-00217-f002:**
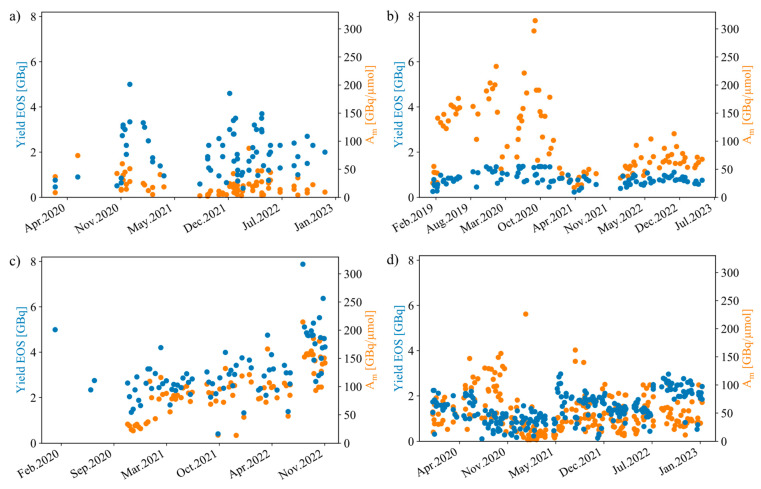
Radioactive yields at EOS (left axis) and molar activities (*A*_m_, right axis) for [^11^C]PiB production as a function of dates at (**a**) AUH, (**b**) HUH, (**c**) OUH, and (**d**) RH. Shows general trend of high yields by use of [^11^C]CO_2_ (**a**,**c**,**d**) and high *A*_m_ by use of [^11^C]CH_4_ (**b**).

**Figure 3 pharmaceuticals-17-00217-f003:**
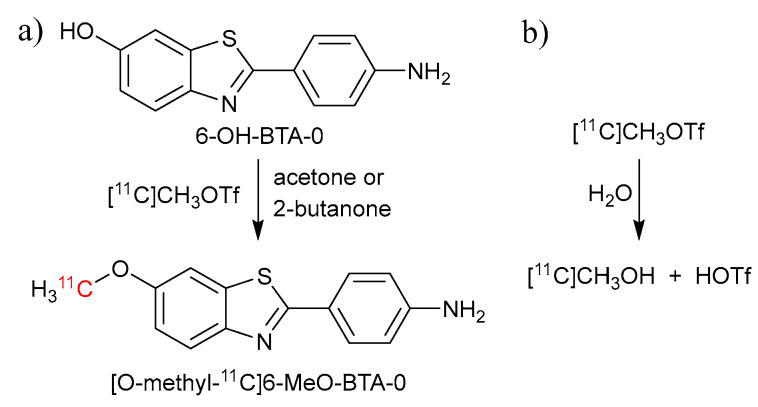
Potential byproducts in the [^11^C]PiB synthesis process. (**a**) O-methylation reaction of the precursor 6-OH-BTA-0 and (**b**) formation of [^11^C]CH_3_OH in the presence of water.

**Figure 4 pharmaceuticals-17-00217-f004:**
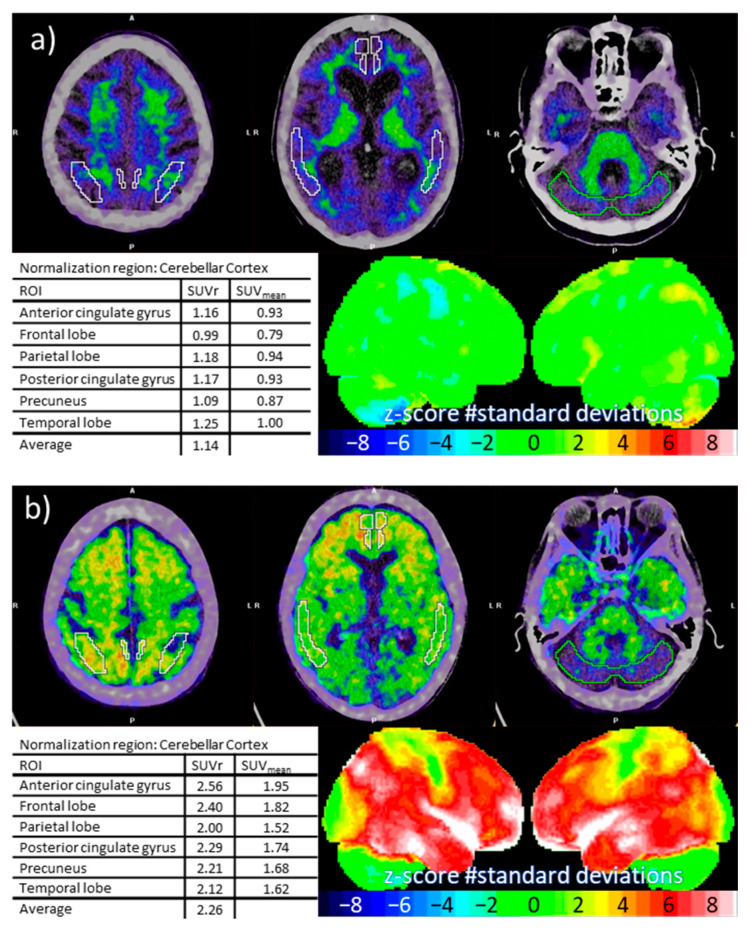
[^11^C]PiB PET scans from two patients suspected of Alzheimer’s disease. A negative scan is seen in (**a**) with VOIs drawn on axial images, corresponding SUV_mean_ ratio (SUVr) with an average < 1.5, and 3D-SSP images (stereotactic surface projections of z-score maps). A positive scan is seen in (**b**) with an average SUV_mean_ ratio > 1.5.

**Table 1 pharmaceuticals-17-00217-t001:** Parameters for [^11^C]PiB production at AUH, HUH, OUH, and RH in the periods from 2020, 2019, 2020, and 2021 until 2023, respectively, where *n* is number of productions in the given periods, *T*_Ni_ is temperature of nickel oven, *T*_Iodine_ is temperature of the iodine oven, *T*_Quartz_ is temperature of quartz oven for iodination reaction, *m* is amount of precursor used, *V*_Sol_ is solvent volume used, *T*_trap_ is trapping temperature of [^11^C]CH_3_OTf in the reaction solvent, *T*_R_ is reaction temperature, *t*_R_ is reaction time, yield is activity measured at EOS, and *A*_m_ is molar activity given at EOS. The AgOTf oven temperature is 190 °C at AUH and 200 °C for the remaining three sites.

	AUH	HUH	OUH	RH
*n*	74	105	84	252
^11^C-precursor	[^11^C]CO_2_ ^a^	[^11^C]CH_4_	[^11^C]CO_2_	[^11^C]CO_2_ ^a^
*T*_Ni_ [°C]	375	N/A	350	350
*T*_Iodine_ [°C]	100	60	100	20–25
*T*_Quartz_ [°C]	740	720	700	760
*m* [mg]	2.0	1.0	1.0	1.0
Solvent	anhydrous acetone	dried 2-butanone	acetone	2-butanone
*V*_Sol_ [μL]	300	150	300	300
*T*_trap_ [°C]	−20	−10	5	−10
*T*_R_ [°C]	50	85	60	80
*t*_R_ [min]	2.5	2.5	2.0	3.0
Yield EOS [GBq]	1.98 ± 1.00	0.83 ± 0.29	3.17 ± 1.20	1.46 ± 0.64
*A*_m_ [GBq/μmol]	21.2 ± 16.8	98.0 ± 61.4	95.6 ± 44.2 ^b^	55.0 ± 50.4

^a^ Use of [^11^C]CH_4_ is possible using the setup; however, [^11^C]CO_2_ is preferred due to higher yields (see main text). ^b^ *A*_m_ was not determined prior to 21 January 2023 due to the experimental setup.

**Table 2 pharmaceuticals-17-00217-t002:** Equipment and parameters for semi-preparative HPLC separation for the [^11^C]PiB production processes at the four sites, AUH, HUH, OUH, and RH. *RT* is retention time, and *V*_frac_ is the volume of the collected [^11^C]PiB fraction.

	AUH	HUH	OUH	RH
Pump(s)	Knauer P 4.1S	2 × Knauer 100 on valve dock	SYKAM S1021	Knauer 100 Knauer P.4.1S
Column	Phenomonex Luna C18, 250 × 10 mm ^a^	Phenomonex Kinetex 2.6 μm C18, 100 Å, 50 × 4.6 mm	Merck Chromolith Performance RP-18e, 100 × 10 mm	Phenomenex Onyx Monolithic C18, 100 × 10 mm
UV detector	Knauer K-2501	Knauer Smartline UV 200	Knauer K-2001	Knauer UV 120 Knauer Azura UVD 215
Gamma detector	TRACERlab FX c built-in	TracerMaker built-in (GM tube)	TRACERlab FX c built-in	Gilson 401 dilutor
Method	Isocratic	Isocratic	Isocratic	Isocratic
Eluent ratio (A:B)	45:55 ^b^	30:70	25:75 ^b^	30:70 ^b^
Eluent A	Ethanol	Ethanol	Ethanol	Ethanol
Eluent B	70 mM NaH_2_PO_4_	0.1% H_3_PO_4_/25 mM ascorbic acid/1% 2-butanone	15 mM H_3_PO_4_	0.1% H_3_PO_4_/25 mM ascorbic acid
Rate (mL/min)	≈6	8	6.5	6
*RT* (min)	7–8	3–4	≈8	5–6
*V*_frac_ [mL]	≈6	2.67	6.5	6

^a^ Further purification of HPLC fraction is performed on a SepPak C18 cartridge (see main text). ^b^ Eluent A and B are mixed in one bottle and fed directly to the HPLC system.

**Table 3 pharmaceuticals-17-00217-t003:** Cyclotron parameters for [^11^C]PiB production at AUH, HUH, OUH, and RH.

	AUH	HUH	OUH	RH
Cyclotron	GE PETtrace 600 ^a^ GE PETtrace 800 ^a^ IBA Cyclone 18/18	IBA Cyclone 18/18	GE PETtrace	CTI XP ^b^ Scanditronix ^c^
Target gas	99.5% N_2_ + 0.5% O_2_ 95% N_2_ + 5% H_2_	95% N_2_ + 5% H_2_	99.5% N_2_ + 0.5% O_2_	99.5% N_2_ + 0.5% O_2_ 95% N_2_ + 5% H_2_
Irradiation time [min]	44.5	25–30	30–60	44.5
Beam current [μA]	60.0 ^a^, 20.0	27.0	60	52.7

^a^ High-pressure targets, used with 60.0 μA beam current. ^b^ For [^11^C]CO_2_ production. ^c^ For [^11^C]CH_4_ production and the cyclotron is from Scanditronix, Uppsala, Sweden.

## Data Availability

Data can be requested by contacting the corresponding author.

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
