# Peer review of "Multicenter Experience with Good Manufacturing Practice Production of [11C]PiB for Amyloid Positron Emission Tomography Imaging"

_pharmaceuticals, 2024, doi:10.3390/ph17020217_

Round 1

Reviewer 1 Report

Comments and Suggestions for Authors

The paper requires major revisions.

1. Can you provide more details on the rationale behind choosing [11C]PiB as the radiopharmaceutical for Alzheimer's disease diagnostics?

2. What specific challenges did you encounter in the [11C]PiB production setups, and how were these challenges addressed?

3. In the comparison of production methods, what factors influenced the preference for [11C]CO2 over [11C]CH4, and vice versa, at different facilities?

4. Could you elaborate on the significance of using different synthesis modules (TracerLab FXC Pro, ScanSys, or TracerMaker) in the [11C]PiB production?

5. How do the results of [11C]PiB production with [11C]CO2 on a TracerLab FX C Pro compare to other synthesis modules in terms of yield and molar activity?

6. Can you discuss the implications of the observed differences in radioactive yield and molar activity on the clinical effectiveness of [11C]PiB PET scans? Add suitable references to the site with [PMID: 37238175, PMID: 37701174]

7. In the synthesis procedures, what led to the selection of specific solvents such as dry acetone or 2-butanone for dissolving the 6-OH-BTA-0 precursor?

8. What considerations influenced the choice of reaction temperature and time for the [11C]PiB production?

9. How does the production method using [11C]CO2 and [11C]CH4 contribute to achieving the desired results in terms of yield and molar activity?

10. Were there any unexpected findings or variations in the [11C]PiB production results among the Danish PET facilities/production sites?

11. Can you provide insights into the specific challenges and potential pitfalls mentioned in the production and use of [11C]PiB?

12. What are the practical implications of the [11C]PiB production setups and experiences discussed in your study for the broader field of Alzheimer's disease diagnostics?

13. How do the synthesis parameters presented in Table 1 contribute to the optimization of [11C]PiB production and its clinical applications?

14. What measures were taken to ensure the reliability and reproducibility of the [11C]PiB production across different sites and synthesis modules?

15. Can you discuss the role of radical scavengers, such as ascorbic acid, in mitigating the radiolytic decomposition process during [11C]PiB synthesis?

16. In the context of the short half-life of 11C, how did you address the challenges related to separation and formulation of [11C]PiB from the reaction mixture?

17. How does the one-step synthesis approach, involving [11C]CH3OTf and the unprotected precursor, compare to the original radiosynthesis method using O-methoxymethyl (MOM) protected precursor?

18. Could you provide more information on the reported one-pot strategies involving catalytic mixtures and their potential for clinical implementation?

19. Considering the evolving landscape of Aβ imaging tracers, how does [11C]PiB compare to 18F-tracers in terms of diagnostic accuracy and practical considerations?

20. What future developments or improvements do you foresee in [11C]PiB production methods, and how might these impact its clinical utilization?

Author Response

Dear Reviewer,
Thank you very much for taking the time to review this manuscript. Please find the detailed responses below and the corresponding revision with track changes in the re-submitted files.

Comments 1: Can you provide more details on the rationale behind choosing [11C]PiB as the radiopharmaceutical for Alzheimer's disease diagnostics?

Response 1: We believe that a detailed rationale for choosing [11C]PiB as the radiopharmaceutical for Alzheimer's disease diagnostics is already described in the abstract, in the introduction p 2, l. 36-45 and at p. 4, l. 131-140. The current golden-standard for diagnosis of AD is [11C]PiB.

Comments 2: What specific challenges did you encounter in the [11C]PiB production setups, and how were these challenges addressed?

Response 2: The challenges that we encounter in the [11C]PiB production setups are described under radiolysis, p. 7, l. 230-258, radiochemical and chemical purity, p. 7, l. 260-286, EtOH content at AUH, p. 8, l. 297-303 and the choice of sterilization filter, p. 9, l. 314-324.

Comments 3: In the comparison of production methods, what factors influenced the preference for [11C]CO2 over [11C]CH4, and vice versa, at different facilities?

Response 3: The cyclotron setup determined the choice. The setup description for HUH p. 4, l. 154-157 is updated. The choice for 11CO2 over 11CH4 at AUH and RH (where the setup allows for both options) is stated at p. 7, l. 230-245.

Comments 4: Could you elaborate on the significance of using different synthesis modules (TracerLab FXC Pro, ScanSys, or TracerMaker) in the [11C]PiB production?

Response 4: Added additional context to introduction p. 4, l. 142-149.2

Comments 5: How do the results of [11C]PiB production with [11C]CO2 on a TracerLab FX C Pro compare to other synthesis modules in terms of yield and molar activity?

Response 5: We believe that the results and comparison of the synthesis modules are summarized in Table 1 and section 2 (p. 5, l. 196-199), and discussed further in section 3 with a summary at p.9 l. 326-341, and conclusion at p. 12 l. 493-510.

Comments 6: Can you discuss the implications of the observed differences in radioactive yield and molar activity on the clinical effectiveness of [11C]PiB PET scans? Add suitable references to the site with [PMID: 37238175, PMID: 37701174]
Response 6: The differences in radioactive yield and molar activity on the clinical effectiveness of [11C]PiB PET scans are described in p. 3 l. 73-93. Various references are already added. We believe that these two suggested references do not fit in here even though they are excellent.

Comments 7: In the synthesis procedures, what led to the selection of specific solvents such as dry acetone or 2-butanone for dissolving the 6-OH-BTA-0 precursor?
Response 7: Selection of the solvents are based on established literature procedures, p. 5, l.171-173.

Comments 8: What considerations influenced the choice of reaction temperature and time for the [11C]PiB production?

Response 8: The temperature is based on the literature procedures where 80 °C is reported for acetone and up-to 75 °C for 2-butanone. Reaction times of 1-2 min are also reported in the literature for reasons stated at p. 5, l. 176-178.

Comments 9: How does the production method using [11C]CO2 and [11C]CH4 contribute to achieving the desired results in terms of yield and molar activity?
Response 9: We believe that this has being addressed under discussion p. 6 l. 219-225 and at p. 9 l. 329-345.

Comments 10: Were there any unexpected findings or variations in the [11C]PiB production results among the Danish PET facilities/production sites?

Response 10: The sudden drop off in molar activity for HUH is a bit unexpected (change of He-purifier does not seem to have helped). Currently under investigation, p. 7, l. 225-228.

Comments 11: Can you provide insights into the specific challenges and potential pitfalls mentioned in the production and use of [11C]PiB?

Response 11: See answers for question 2. As well as the general considerations regarding 11C chemistry in the introduction esp. p. 3, l. 66-93.

Comments 12: What are the practical implications of the [11C]PiB production setups and experiences discussed in your study for the broader field of Alzheimer's disease diagnostics?
Response 12: Described at p. 3, l. 68-70.

Comments 13: How do the synthesis parameters presented in Table 1 contribute to the optimization of [11C]PiB production and its clinical applications?
Response 13: It serves as starting point for other sites who wishes to implement PIB production of their own. Added additional sentences in abstract (p. 1, l. 30-31), introduction (p. 4, l.147-148), and conclusion (p. 13, l. 495-497).

Comments 14: What measures were taken to ensure the reliability and reproducibility of the [11C]PiB production across different sites and synthesis modules?
Response 14: This is already answered in question 2, which implies a push towards reliability and reproducibility, as well as minimizing sources of carbon-12, p. 7, l. 222-223.

Comments 15: Can you discuss the role of radical scavengers, such as ascorbic acid, in mitigating the radiolytic decomposition process during [11C]PiB synthesis?
Response 15: The role of radical scavengers is described at p. 3, l. 86-90, and briefly discussed in present work how the yield was improved, p. 7, l. 248-258. This is due to antioxidant property, added sentence and reference to literature describing this property p. 3, l. 88-89.

Comments 16: In the context of the short half-life of 11C, how did you address the challenges related to separation and formulation of [11C]PiB from the reaction mixture?
Response 16: The separation and formulation are by semi-prep. HPLC. This is described at p. 4, l. 115-120 and p. 5, l. 190-191.

Comments 17: How does the one-step synthesis approach, involving [11C]CH3OTf and the unprotected precursor, compare to the original radiosynthesis method using O-methoxymethyl (MOM) protected precursor?
Response 17: The comparison is added to the manuscript (see p. 3, l. 96-98).

Comments 18: Could you provide more information on the reported one-pot strategies involving catalytic mixtures and their potential for clinical implementation?
Response 18: This is beyond the scope of the presented work. We have added a sentence to direct the interested reader to the relevant referenced literature p. 3, l. 113.

Comments 19: Considering the evolving landscape of Aβ imaging tracers, how does [11C]PiB compare to 18F-tracers in terms of diagnostic accuracy and practical considerations?
Response 19: This is described at p. 4, l. 132-134.

Comments 20: What future developments or improvements do you foresee in [11C]PiB production methods, and how might these impact its clinical utilization?
Response 20: Future development for [11C]PiB production should be cassette based methods for easier production (p. 4, l. 122-126).

Kind regards,

Tri Hien Viet Huynh

Reviewer 2 Report

Comments and Suggestions for Authors

In the article by Anders B. A. Andersen et al. two [11C]PiB production methods are described based on r [11C]CO2 or [11C]CH4 and compared with an assessment of suitability for practical use. Using of [11C]CO2 as precursor produces significantly higher radioactive yields (2.5 times higher) compared to [11C]CH4. Three different types of synthesis modules are used for the [11C]PiB production (TracerLab FXC Pro, ScanSys, or TracerMaker) from either of the two 11C-precursors, and variations in the synthesis conditions as well as the purification conditions of the [11C]PiB product was carried out. The described synthetic protocols can be very useful for researchersб as well as for medical practitioners. I recommend the publication of this article.

Comments on the Quality of English Language

Minor editing of English language may be required

Author Response

Dear Reviewer,

Thank you very much for taking the time to review this manuscript. Please find the detailed responses below and the corresponding revision with track changes in the re-submitted files. Thank you for recommending publication of this article.

Comments 1: Minor editing of English language may be required.

Response 1: We have performed second round of proofreading.

Kind regards,
Tri Hien Viet Huynh, Ph.D.
Associate Professor
(Head of Cyclotron and Radiochemistry-QP)
Department of Nuclear Medicine
Herlev Hospital, University of Copenhagen
Borgmester Ib Juuls vej 31, 3. etage
2730 Herlev, Denmark
Tlf: +45 38681112

Reviewer 3 Report

Comments and Suggestions for Authors

   - The explanation of the pathways in Figure 1 is not detailed enough. It would be helpful to provide a more thorough description of the key steps in each pathway.

   - Clarify the abbreviations used in the figure, such as "TNi," "TIodine," etc.

   - The table includes various parameters for [11C]PiB production but lacks clear headings and units for each parameter. 

   - There are inconsistencies in the formatting of the table. Ensure consistent formatting for all entries.

   - The discussion on the comparison between [11C]CO2 and [11C]CH4 methods lacks depth. Provide more insight into the advantages and disadvantages of each method.

   - Discuss the significance of the observed differences in radioactive yields and molar activities for [11C]PiB.

   - Improve the labeling of the axes in Figure 2 for better readability. Clearly indicate what the values on the axes represent.

   - Provide a brief caption for Figure 2 to explain its purpose and what the reader should take away from it.

  - Expand the explanations in Figure 3 to include details about the reaction conditions and the significance of potential by-products. Provide more context for the reader.

   - The manuscript lacks subheadings within sections, making it challenging to follow the flow of the paper. Consider adding subheadings to enhance the organization.

   - Ensure consistency in terminology and abbreviations throughout the manuscript.

Comments on the Quality of English Language

There are many long lines 

Author Response

Dear Reviewer,

Thank you very much for taking the time to review this manuscript. Please find the detailed responses below and the corresponding revision with track changes in the re-submitted files.

Comments 1: The explanation of the pathways in Figure 1 is not detailed enough. It would be helpful to provide a more thorough description of the key steps in each pathway.
Response 1: Thank you for pointing this out. We see that the explanation was mostly done in the figure text. Added description in the main text p. 3, l. 98-107 as well as additional information to Fig 1. Moreover, the figure text is also referring to table 1 with the specific synthesis parameters for each site.

Comments 2: Clarify the abbreviations used in the figure, such as "TNi," "TIodine," etc.
Response 2: I think table 1 is meant and not figure (1?). The abbreviations are specified in the table text, p. 2, l. 57-63.

Comments 3: The table includes various parameters for [11C]PiB production but lacks clear headings and units for each parameter.
Response 3: Table 1? Units are specified in the parameter column (far-left column). Changed the formatting to bold for the parameters to indicate they are “headings” in all three tables.

Comments 4: There are inconsistencies in the formatting of the table. Ensure consistent formatting for all entries.
Response 4: The inconsistencies in range vs. specific value due to different production methods at the different production sites. The difference in significant decimals is due to different precision in e.g., measurement of temperature and mass of precursor. Hence, it’s impossible for the formatting to be consistent with the presented data.

Comments 5: The discussion on the comparison between [11C]CO2 and [11C]CH4 methods lacks depth. Provide more insight into the advantages and disadvantages of each method
Response 5: This depends on cyclotron targets and production setups, but overall trend is CO2→ high radioactive yield, and CH4 → high molar activity (Am). High Am is desirable for PET brain scans (see answer to question below). But high Am also cause problems with product purity. All of this discussed in length at p. 6, l.208-228. Moreover, all presented methods produce PIB in yields and Am suitable for clinical use.

Comments 6: Discuss the significance of the observed differences in radioactive yields and molar activities for [11C]PiB.
Response 6: High molar activity is usually important for brain PET-scans, introduced at p. 3, l. 79-84. Added context/reference to the reader in the discussion section (p. 7, l. 230-231) to better clarify this.

Comments 7: Improve the labeling of the axes in Figure 2 for better readability. Clearly indicate what the values on the axes represent.
Response 7: This is improved. Units are defined on the left- and right-axis and is further defined in the figure text for clarity. Rename the x-axis in the figure 2 for clarification.

Comments 8: Provide a brief caption for Figure 2 to explain its purpose and what the reader should take away from it.
Response 8: We have added a short “take-home”-message to the figure 2 text.

Comments 9: Expand the explanations in Figure 3 to include details about the reaction conditions and the significance of potential by-products. Provide more context for the reader.
Response 9: This is already described in the main-text p. 7, l. 260-274. The significance is related to GMP-production of pharmaceuticals for human use, where impurities are limited by product specifications introduced at p. 7, l. 235-238 and p. 8 l. 280-286.

Comments 10: The manuscript lacks subheadings within sections, making it challenging to follow the flow of the paper. Consider adding subheadings to enhance the organization.
Response 10: Thank you for this suggestion. We have added subheadings to the discussion section to promote readability.

Comments 11: Ensure consistency in terminology and abbreviations throughout the manuscript.
Response 11: We have performed second round of proofreading.

12. Response to Comments on the Quality of English Language
Response 12: We have performed second round of proofreading and shortened some of the long lines.

Kind regards,

Tri Hien Viet Huynh

Reviewer 4 Report

Comments and Suggestions for Authors

This is an interesting article about a relatively new PET tracer, describing its synthesis as performed at several hospitals in Denmark. The methods and results are clearly described. The discussion adequately determines the implications of the techniques used and also includes a nice section about clinical applications. No scientific inaccuracies detected. No obvious areas in need of improvement.

Author Response

Dear Reviewer,

Thank you very much for taking the time to review this manuscript. Great, that the manuscript is accepted without further comments.
We wish you a good day.

Kind regards,
Tri Hien Viet Huynh, Ph.D.
Associate Professor
(Head of Cyclotron and Radiochemistry-QP)
Department of Nuclear Medicine
Herlev Hospital, University of Copenhagen
Borgmester Ib Juuls vej 31, 3. etage
2730 Herlev, Denmark

Reviewer 5 Report

Comments and Suggestions for Authors

The manuscript by Andersen et al. “Multicenter experience with GMP production of [11C]PiB for amyloid PET imaging“ describes variations in productions methods of [11C]-labeled drugs for PET imaging. The synthesis of radiopharmaceuticals is extremely important and challenging task. Careful evaluation of synthetic methodology allows to find all the pros and cons of each reaction setup.  Author were able to choose the most preferable production method of [11C]-drugs. In my opinion this manuscript is suitable for publication in Pharmaceuticals after minor revision.

Abbreviations AUH, HUH, OUH, and RH were not explained in the manuscript (or I missed it).

Author Response

Dear Reviewer,

Thank you very much for taking the time to review this manuscript. Please find the detailed responses below and the corresponding revision with track changes in the re-submitted files.

Comments 1: Abbreviations AUH, HUH, OUH, and RH were not explained in the manuscript (or I missed it).

Response 1: They are defined in the main text at the end of the introduction p. 4, l. 137-140?

Kind regards,
Tri Hien Viet Huynh, Ph.D.
Associate Professor
(Head of Cyclotron and Radiochemistry-QP)
Department of Nuclear Medicine
Herlev Hospital, University of Copenhagen
Borgmester Ib Juuls vej 31, 3. etage
2730 Herlev, Denmark

Round 2

Reviewer 1 Report

Comments and Suggestions for Authors

The article is publishable; all comments have been addressed.